# Machine Learning Method to Explore the Correlation between Fly Ash Content and Chloride Resistance

**DOI:** 10.3390/ma17051192

**Published:** 2024-03-04

**Authors:** Ruiqi Wang, Yupeng Huo, Teng Wang, Peng Hou, Zuo Gong, Guodong Li, Changyan Li

**Affiliations:** 1College of Transportation, Inner Mongolia University, Hohhot 010031, China; 32015127@mail.imu.edu.cn (R.W.); 32124006@mail.imu.edu.cn (Y.H.); 32124046@mail.imu.edu.cn (T.W.); 2College of Chemistry and Chemical Engineering, Inner Mongolia University, Hohhot 010031, China; 32207133@mail.imu.edu.cn (P.H.); 32207131@mail.imu.edu.cn (Z.G.)

**Keywords:** durability of concrete, fly ash concrete, fly ash dosage, machine learning, chlorine resistance

## Abstract

Chloride ion corrosion has been considered to be one of the main reasons for durability deterioration of reinforced concrete structures in marine or chlorine-containing deicing salt environments. This paper studies the relationship between the amount of fly ash and the durability of concrete, especially the resistance to chloride ion erosion. The heat trend map of total chloride ion factor correlation displayed that the ranking of factor correlations was as follows: sampling depth > cement dosage > fly ash dosage. In order to verify the effect of fly ash dosage on chloride ion resistance, three different machine learning algorithms (RF, GBR, DT) are employed to predict the total chloride content of fly ash proportioned concrete with varying admixture ratios, which are evaluated based on *R*^2^, *MSE*, *RMSE*, and *MAE*. The results predicted by the RF model show that the threshold of fly ash admixture in chlorinated salt environments is 30–40%. Replacing part of cement with fly ash in the mixture of concrete below this threshold of fly ash, it could change the phase structure and pore structure, which could improve the permeability of fly ash concrete and reduce the content of free chloride ions in the system. Machine learning modeling using sample data can accurately predict concrete properties, which effectively reduce engineering tests. The development of machine learning models is essential for the decarbonization and intelligence of engineering.

## 1. Introduction

Reinforced concrete structures are extensively utilized in coastal and marine engineering projects. In such environments, the durability of these structures is known to deteriorate primarily due to chloride ion corrosion induced by chlorinated deicing salt or marine conditions. Conventionally, a protective layer is applied to the reinforcement surface within the concrete to prevent corrosion. However, the protective layer deteriorates and delaminates when the concentration of chloride ions reaches a certain threshold, resulting in reinforcement corrosion and compromising the structural safety, serviceability, and durability [1,2,3,4,5,6,7]. Hence, investigating the chloride transport processes within concrete structures is vital for ensuring their long-term durability and service life.

Gesoǧlu [8,9] discovered that a fly ash (FA) content of 60% reduced the water permeability of fly ash concrete by 50% compared to regular concrete. Khatib et al. [10] observed that the water absorption of fly ash concrete with a high admixture is dependent on age. They found that extending the concrete age from 56 to 130 days with an 80% fly ash admixture reduced water absorption by 2%. Islam et al. [11] conducted a study on the permeability of concrete with fly ash admixture and determined that the hydration of fly ash concrete generates C-S-H gel, which enhances structural denseness. They found that the permeability of concrete with 40% fly ash admixture was 26% lower than that of normal concrete and increasing the FA content to 60% resulted in a 21% decrease in the permeability coefficient. Burden [12] investigated the relationship between the amount of fly ash admixture in concrete and carbonation resistance. They observed that with fly ash admixture levels up to 30%, 40%, and 50%, the carbonation rate of fly ash concrete was higher than that of normal concrete. Turk’s [9] study revealed a gradual decrease in carbonation resistance when the FA content in the concrete reached 40%. Kurda et al. [13] conducted a study on the influence of microstructure on concrete durability when combining recycled aggregate concrete (RCA) with fly ash (FA). They found that the presence of Ca(OH)_2_ in RCA compensates for the Ca(OH)_2_ consumed by SiO_2_ in FA. As carbonation time increases, the volcanic ash reaction produces a C-S-H gel, which fills the pores and enhances the concrete’s density. This process improves the resistance of concrete to CO_2_ intrusion and enhances its carbonation capacity. Merida et al. [14] discovered that increased incorporation of fly ash promotes resistance to sulfate attack and enhances the volcanic ash reaction, resulting in more C-S-H production. Liu [15] also observed a significant improvement in concrete’s resistance to sulfate attack with increased fly ash incorporation, with optimal performance achieved at 40% incorporation. Zobal [16] demonstrated a negative correlation between the dynamic modulus of concrete and the number of freeze–thaw cycles, as well as fly ash incorporation. After 50 cycles, the control concrete with a flexural strength of 5 MPa exhibited superior performance compared to the 1.2 MPa strength of concrete with a 50% fly ash admixture. Ramesh [17] investigated fly ash mortar and found that after 300 freeze–thaw cycles, the compressive strength of mortar with a 10% fly ash admixture reduced by 30.7% compared to the control sample. Moreover, concrete containing fly ash displayed less resistance to freeze–thaw damage. Shaikh [18] observed lower corrosion mass loss of reinforcement in concrete with a 40% fly ash admixture. Jiang [19] determined that the addition of fly ash enhances the corrosion resistance of concrete, particularly in large admixture fly ash concrete samples, which effectively resisted corrosion at 56 and 118 days of age compared to the control samples. Chung [20] identified a relationship between the volcanic ash reaction of fly ash and corrosion resistance. The volcanic ash reaction reduces the presence of Ca(OH)_2_, decreases passivation in reinforced concrete, and generates additional secondary cementitious material that fills the pores of concrete. This process improves corrosion resistance by reducing permeability.

Current research on the durability of fly ash concrete primarily focuses on water permeability, carbonation resistance, sulfate resistance, freeze–thaw resistance, and chloride ion corrosion and corrosion resistance. Moreover, the majority of studies investigating the durability and service life of concrete structures rely on the preparation of standard concrete specimens and prolonged experimental testing, often with limitations. In coarse engineering, there has been a growing interest in employing machine learning techniques, which utilize sample data for rational analysis, to predict key properties of concrete such as compressive strength, flowability, durability, and carbon footprint. This approach effectively enhances efficiency in both laboratory and coarse engineering settings. Ahmad [21] and Song [22] employed artificial neural network (ANN) and decision tree (DT) models to evaluate 60 samples of ceramic waste cement dosage, water–cement ratio, and other factors, ultimately predicting their compressive strengths. The results revealed that the artificial neural network approach produced more satisfactory regression values for the strength prediction model of ceramic solid waste concrete.

Nguyen [23] and Kaloop [24], on the other hand, employed gradient-tree boosting regression (GBR), extreme gradient boosting (XGBoost), Support Vector Machine (SVM), and Multilayer Perceptron (MLP) models to predict the correlation between compressive strength and tensile strength of high-performance concrete (HPC) using 1133 samples and considering factors such as mix ratio. The findings indicated that the age, cement quantity, and coarse aggregate quantity of HPC exert a significant influence on its mechanical properties. The predicted values generated by the GBR algorithm and XGBoost algorithm closely aligned with the experimental results, demonstrating the potential of these methods for predicting the mechanical properties of high-performance concrete. Bajpai [25] employed artificial neural network (ANN), decision tree (DT), and linear regression (LR) algorithms to predict carbon emissions based on various mix ratios of concrete. The study found that the highest reduction in carbon emissions occurred when fly ash geopolymer was prepared by substituting water glass with silica fume. The ANN algorithm exhibited the best prediction performance, while LR showed the poorest performance. Lien [26] and Liu [27] utilized emotional neural network-chaotic particle swarm optimization (EANN-CPSO), first-principles molecular dynamics (FPMD), and conventional linear regression (LR) algorithms to predict carbon emissions for 60 samples of dense green concrete (SCGC). Their goal was to analyze the mechanical properties using predictive models. The results demonstrated that hybrid models and optimization methods, combined with additional models, could improve the accuracy of predictions. Liu [28,29,30] employed artificial neural networks (ANN) to predict chloride diffusion coefficients in 112 mix ratios of recycled concrete. The study identified the water–cement ratio and fine aggregate as sensitive factors affecting the predictions. Based on performance parameter values, the ANN model exhibited better generalization capabilities.

The innovation of this paper is to propose a method to analyze the chlorine resistance of fly ash-based materials using machine learning, to compare the predictive performance of various models for the total chloride content of concrete, and to analyze and discuss the mechanism of the evolution of the performance of fly ash-based materials in the presence of chloride ions. Firstly, we explore the relationship between the amount of fly ash and the durability of concrete and obtain the heat trend map. Secondly, the influence factor is ranked according to the heat map. At the same time, the evolution mechanism of resistance to chloride ion erosion of fly ash concrete is analyzed. Finally, the correlation between fly ash admixture and concrete durability is evaluated using three different machine learning (RF, GBR, DT) models. This project will be of benefit in finding methods for improving chloride ion penetration, and to avoid the deterioration of the durability of reinforced concrete structures in marine or chlorine-containing deicing salt environments.

## 2. Method

Three machine learning models are employed, namely, random forest (RF), gradient boosting regression (GBR) [31,32,33,34], and decision tree (DT) [35,36,37] (shown in Figure 1, Figure 2 and Figure 3 respectively). To further assess and compare the combined prediction performance of these models, several statistical methods are utilized. These methods evaluate the performance of each model by considering various metrics, including the coefficient of determination (*R*^2^), mean square error (*MSE*), root mean square error (*RMSE*), and mean absolute error (*MAE*). These parameters not only quantify the accuracy of individual algorithms but also facilitate comparisons between different models, helping identify the most suitable and applicable model for a given database. The equations representing these evaluation metrics are provided in the equations section (Appendix A) [38,39,40,41].

## 3. Result Analysis and Discussion

### 3.1. Analysis of Key Influencing Factors on Chloride Ion Permeability Resistance of Concrete

Chloride ion resistance is one of the factors affecting the durability performance of concrete. The influence of chloride ions on concrete durability is mainly reflected in that its invasion into concrete will destroy the passivation film on the surface of reinforced concrete, which in turn induces corrosion of reinforced concrete. The correlation analysis of 503 sets of data is carried out in this paper. All these data come from Sun’s doctoral thesis [42].

An’s research [43] found that 30% fly ash instead of cement can improve the permeability of concrete, and the concentration of free chloride ions is related to the sampling depth. Wang [44] studied cementitious materials with less than 30% fly ash and found that chlorine resistance was mainly determined by permeability, which reduced the chloride diffusion coefficient in the system to 24%. This is similar to the results found by Cheewaket’s [45] and Liu’s [46] teams. In addition, a large amount of fly ash also has attracted the attention of researchers. Kayali and his coworkers [47] focused on cementitious materials with ≤70% fly ash with 10% silica fume replacing cement by rapid chloride ion penetration testing. The results exhibited that the best chlorine resistance and the least degree of chloride corrosion was achieved with 50% fly ash, which increases strength and durability. At the same time, Rumman’s [48] team also found a positive correlation between fly ash admixture and chlorine resistance of concrete, when 20~35% fly ash was incorporated in concrete. It was well known that the incorporation of fly ash could fill the pores of concrete, which can resist the diffusion of CO_2_ in concrete and inhibit the dehydrogenation of steel bars. Thereby, the incorporation of fly ash can change the carbonization resistance of concrete. Salcedo [49] and his colleagues found that the chlorine resistance of concrete was significantly improved while the carbonization resistance was significantly decreased when the amount of fly ash incorporation reaches 20~30%. Huang [50] found that the difference between the interfacial transition zone (ITZ) and matrix migration coefficients gradually decreased with an increasing in fly ash when 0–50% content of fly ash replaces cement. In fact, there are certain limitations in using these test methods to verify the performance of concrete structures. Recently, Yu [51] developed a prediction model for the chloride diffusion coefficient of concrete and found that the factor chloride diffusion coefficient was positively correlated with the water–cement ratio and negatively correlated with the content of fly ash. This kind of new material development mode supplies us with a good idea for analyzing the chlorine resistance, which could shorten the development cycle and reduce the cost. In a word, when a certain amount of fly ash is incorporated into cement, it would change the porosity and colloidal phase structure, which results in decreasing the permeability of chloride ions and increasing the diffusion resistance of chloride ions. At the same time, it improves the carbonation resistance and enhances the strength and durability of concrete (shown in Figure 4).

### 3.2. Heat Trend Map Analysis of the Correlation of Influencing Factors

There are many factors affecting the chloride ion resistance of fly ash-based concrete. This study was conducted in order to find the relationship between these factors and chloride ion resistance. It was observed that the correlation coefficients of coarse aggregate and soaking time are 0.02 and 0.017, respectively, which indicated that there is a relatively weak correlation between coarse aggregate and soaking time with the total chloride ion content. Meanwhile, sampling depth exhibited the strongest correlation with a coefficient of −0.69. These findings align with real-world knowledge. In addition, the heat trend map of total chloride ion factor correlation displayed that the ranking of factor correlations was as follows: sampling depth > cement dosage > fly ash dosage. The total chloride content at different sampling depths can be directly attributed to variations in chloride permeability within the concrete. The chemical binding of chlorides to C_3_A and C_4_AF generated during the hydration of cement in the fly ash–cement system can influence the chloride adsorption capacity of fly ash concrete. Additionally, fly ash, characterized by its high alumina content, reacts with chlorides to form chloroaluminate hydrates (Friedel’s salt) [52]. Moreover, it physically fills microscopic pores and leads to a series of changes including a denser pore structure, refined pores, optimized pore size distribution, reduced critical pore size, and minimum pore size connecting the pores. As a result, pore connectivity and ion mobility are effectively reduced, thus decreasing chloride ion permeability. Consequently, the total chloride ion content in fly ash concrete is reduced. The final results were found to be consistent with the experimental measured changes. The entry of chloride ions into concrete is very closely related to the improvement of the pore structure and the gel phase structure of fly ash incorporated into concrete.

### 3.3. Microstructural and Pore Structure Analysis of the Deterioration Mechanism of Concrete Resistance to Chloride Ions

Based on the analysis of Figure 5, it clearly shows the correlation of influencing factors for total chloride ion. The addition of fly ash has an important effect on the phase structure and pore structure of concrete cementing material. As the percentage of fly ash content increases, the hydration process generates more hydrated calcium silicate (C-S-H) gels, which can physically adsorb a greater amount of chloride. Some of the chloride ions can also chemically bind to C_3_A and C_4_AF, as shown in Figure 6, produced during cement hydration, thereby improving the chloride adsorption capacity in fly ash concrete systems. Additionally, an increase in fly ash admixture leads to a higher alumina content in the concrete. The reaction between alumina and chloride produces chloroaluminate hydrate, commonly known as Friedel’s salt [45,52,53,54,55,56,57,58,59] shown in Figure 7 and Figure 8. Consequently, increasing the fly ash admixture and the inclusion of Al_2_O_3_ effectively enhance the chloride binding capacity of fly ash concrete and slow down the degradation caused by material corrosion. The surface hydration products (C_3_A, C_4_AF, C-S-H) bind to free chloride [60], effectively controlling the reduction of free chloride content in fly ash concrete systems [61,62].

The reason why the chloride resistance of fly ash concrete becomes more prominent in later stages is due to the presence of dense spherical vitreous microspheres on the surface of fly ash particles. These microspheres initially hinder the hydration process, and it is only during the later stages when the vitreous surface has undergone significant erosion that its hydration products can combine with the free chlorides in the system. In conclusion, in the hydration system of fly ash–cement concrete, the increase in fly ash admixture enhances both the physical adsorption and chemical reaction capabilities of chlorides. This dual enhancement contributes to the improved chloride resistance of fly ash concrete. Since the variation in fly ash admixture corresponds to changes in material ratios, optimizing the ratio represents the most direct and effective approach to enhancing the chloride resistance of fly ash concrete.

To improve the chloride resistance of fly ash concrete, the primary approach is to reduce permeability and optimize the pore structure [64,65,66]. Optimizing the pore structure emerges as the most effective method. With an increase in fly ash admixture from 0% to 27%, the porosity decreases from 40.59% to 27.28% (Table 1), resulting in a more uniform and rational distribution of pores (shown in Figure 7 and Figure 9). This outcome can be attributed to the volcanic ash reaction of fly ash, which consumes excess Ca (OH)_2_ (CH) and promotes the formation of hydrated calcium silicate gel. Additionally, the amorphous C-S-H gel with calcium alumina undergoes continuous crystallization, growth, diffusion, and cross-linking, contributing to the development of a networked microstructure. Wang [67] also observed that fly ash refines the pore structure and reduces pore size, with the rate of refinement increasing with age. Sadrmomtazi [68] and Hassen [69] found that the addition of 20% fly ash admixture resulted in a 20.9% and 16.1% reduction in porosity at 3 days compared to 28 and 90 days, respectively. This reduction in porosity improves the pore structure and permeability. Additionally, unreacted fly ash particles tend to accumulate, as shown in Figure 7, physically filling microscopic pores and further densifying the pore structure. This optimization of pore size distribution reduces critical pore size and the minimum pore size of connecting pores, which effectively decreases pore connectivity, ion mobility, and chloride ion permeability. As a result, the ion transport rate within the system is reduced, providing resistance to chloride ion permeation and achieving a “low permeability” grade for fly ash concrete. Consequently, the chloride resistance of fly ash concrete is enhanced. Furthermore, in addition to increasing volcanic ash activity, the inclusion of fly ash particles fills the void spaces in the cementitious matrix, creating a microaggregate effect and improving concrete durability [70]. However, a higher fly ash admixture can lead to reduced adhesion capacity of fly ash particles, resulting in their accumulation without proper hydration. This may increase concrete porosity, decrease the alkalinity of the pore solution, and potentially release previously bound chlorides, exacerbating chloride-induced deterioration and reducing the service life of the material.

### 3.4. Machine Learning Method to Explore the Analysis of Fly Ash Admixture with Durability and Chloride Resistance

#### 3.4.1. Model Data for the Sample

Sun [42] conducted a comprehensive investigation on the durability and life prediction of chloride ion-eroded concrete. This paper focused on validating the total chloride ion content of fly ash concrete after exposure to chloride ions. Based on this work, we compiled a dataset for modeling purposes, comprising a total of 503 concrete mixtures with varying fly ash content and their corresponding total chloride ion contents. The dataset considered several factors, including cement dosage, fly ash dosage, coarse and fine aggregate dosage, age, soaking time, sampling depth, and water–cement ratio. All input variables were treated as independent factors. While additional material properties such as particle size distribution were involved in the analysis, the impact of aggregate size distribution on strength was not considered due to the challenge of quantifying it as a parameter. For the purpose of machine learning prediction in this chapter, we selected seven factors as input variables including cement dosage, fly ash dosage, coarse and fine aggregate dosage, age, soaking time, sampling depth, and water–cement ratio. The total chloride ion content, which best reflects the degree of chloride ion attack on the material, was chosen as the output variable. The dataset was divided into 70% for training the model and 30% for testing. We selected the RF, GBR, and DT models with the highest coefficients of determination (*R*^2^) to compare their prediction performance. The statistical parameters of the model dataset are presented in Table 2.

The dataset consisted of 503 samples with eight variable factors. Of the data, 70% was used for training the models, while the remaining 30% was utilized for testing. Among the input factors, sampling depth, cement dosage, and fly ash dosage exerted the most significant influence on the output results, while water–cement ratio, age, soaking time, and coarse and fine aggregates had a lesser impact.

#### 3.4.2. Training and Testing of Machine Learning Models

From Figure 10 and Appendix A, the prediction outcomes of the total chloride ion content for three distinct machine learning models can be observed. In each graph, the blue points represent the predicted values generated by the models, while the red points represent the measured values. The black histograms at the bottom of the graphs depict the corresponding error values between the predicted and the measured values. The data results in Figure 10 and Appendix A show the error between the actual and predicted values. Both the GBR and DT algorithms display errors within 0.05%. Upon analyzing the test set, it is observed that some prediction points generated by GBR and DT deviate more from the actual total chloride content values.

Based on the evaluation parameters presented in Table 3 and Figure 5, a consistent trend of variation was observed among the three models. The analysis revealed that the GBR model exhibited the highest accuracy in terms of *R*^2^ when considering the entire database as a reference value. However, when comparing the prediction performance of the different models, it was found that the *R*^2^ of the RF training set was 0.9887, slightly surpassing that of the test set. On the other hand, the *R*^2^ values for the test sets of the GBR and DT algorithms were 0.8859 and 0.8632, respectively, considerably lower than their respective training sets. The low error values of *MSE* and *MAE* proved that RF is a good model to predict total chloride ion content [71,72]. This indicates that GBR and DT models lacked good generalization ability and prediction stability, despite being able to predict functional relationships within a given database. This overfitting phenomenon was further validated by other evaluation parameters such as *MSE*, *RMSE*, and *MAE*. In comparison, RF demonstrated a more stable prediction accuracy among the three methods.

#### 3.4.3. Error Range and Percentage Distribution of Machine Learning Models

Each machine learning model exhibits a few abnormal sample points, while the remaining samples are uniformly distributed on both sides of the *y = x* function, indicating a relatively accurate prediction of total chloride ion content in fly ash concrete by the machine learning models. The blue dots represent training data, while the red dots represent testing data. The amount of scatter points of the testing data distributed far away from the 45° dotted line indicates the accuracy and applicability of the models changing significantly under different conditions [72]. RF displays the highest number of samples within the −20% to 20% error range, signifying a more concentrated data distribution among the three machine learning algorithms. Additionally, from Figure 11 and Appendix A, it can be observed that while DT has a slightly higher number of samples within the −20% to 20% range compared to RF, GBR demonstrates prediction results that are closer to the test results, with sample points more concentrated near the reference line shown in the figure. In Appendix A, the test values of the DT model contain more points outside the ±20% error range, indicating lower accuracy when performing performance predictions in the test group. Upon considering all comparisons, it is evident that the RF model achieves higher prediction accuracy.

Figure 12 and Appendix A present the error percentages associated with different machine learning algorithms and their respective data within the corresponding error ranges. The fitted curve equations Equation (1) and Appendix A are also provided. Utilizing relative error values allows for a better reflection of the confidence in the model’s prediction performance. Hence, percentage error values are employed for fitting the normal distribution, as they help avoid situations where the performance of total chloride ion content deviates significantly. The distribution of the fitted results and percentages demonstrates that all three machine learning algorithms converge towards zero [73].
(1)$$y=0.00774+\frac{0.08877}{0.16738}e^{-2{\left(\frac{x-0.0035}{0.16738}\right)}^{2}}$$

The prediction performance of the three models was compared, and it was found that RF, GBR, and DT all exhibited strong prediction results with the predicted values closely aligning with the actual values. The majority of errors fell within a range of 20% (Figure 11 and Appendix A). In terms of *R*^2^, the three algorithms ranked as follows in descending order: RF > GBR > DT. Notably, the random forest algorithm (RF) demonstrated the most stable prediction ability when considering other characteristic parameters. When compared to the other two models, the error percentage range of RF appeared to be more reasonable.

This paper pioneers the use of machine learning models, algorithms, and error rates in analyzing the relationship between the amount of fly ash and the durability of concrete, especially the resistance to chloride ion erosion. It not only overcomes the uncertainty of blindly selecting but also effectively reduces the massive loss of human and material resources in engineering. However, the accuracy of machine learning methods depends on the representativeness and quantity of data. It can be obviously found that DT models are prone to overfitting, GBR are less sensitive to abnormal value, and RF models have higher randomness, which results in a certain limitation in practical work. In future work, we think more sample parameters and related factors can be introduced as dependent variables, especially the microscopic parameters of fly ash concrete’s resistance to chloride ion erosion, which will provide an important and valuable reference value for artificial intelligence and green low-carbon civil engineering.

## 4. Conclusions

Chloride ion corrosion is considered to be one of the main causes of durability deterioration of reinforced concrete structures in the marine environment. In this paper, a heat trend diagram is used to analyze the correlation between total chloride ions and the influencing factors. Three different machine learning algorithms (RF, GBR, DT) are employed to predict the total chloride content of fly ash proportioned concrete with varying admixture ratios, aiming to enhance its resistance against chloride ion penetration. The following conclusions were derived from the results:
According to the results of the heat diagram, sampling depth is negatively correlated and has the greatest effect on total chloride ions. The threshold range for the fly ash admixture is around 30–40%. In the hydration system of fly ash–cement concrete, the presence of fly ash admixture enhances the binding capacity, adsorption capacity, and chemical reaction capacity of chloride. More importantly, the doping of fly ash can change the pore structure and the phase structure of the cementing material, thereby effectively improving the chloride resistance of fly ash concrete.The models’ predictive performance was evaluated based on *R*^2^, *MSE*, *RMSE*, and *MAE*. All three models (RF, GBR, DT) exhibited favorable accuracy and stability, with predicted values closely aligning with actual values and errors typically within a 20% range. The descending order of *R*^2^ values for the algorithms was RF > GBR > DT, which suggests that GBR and DT have relatively weaker generalization abilities. Compared to the other two models, RF demonstrated superior robustness, as indicated by error percentages within the −20% to 20% range and a steep normal distribution fitting curve. The random forest algorithm (RF) displayed the most stable prediction capability.


## Figures and Tables

**Figure 1 materials-17-01192-f001:**
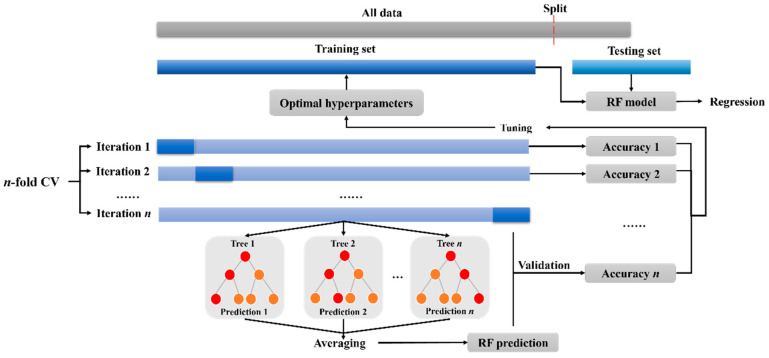
Random forest modelling framework.

**Figure 2 materials-17-01192-f002:**
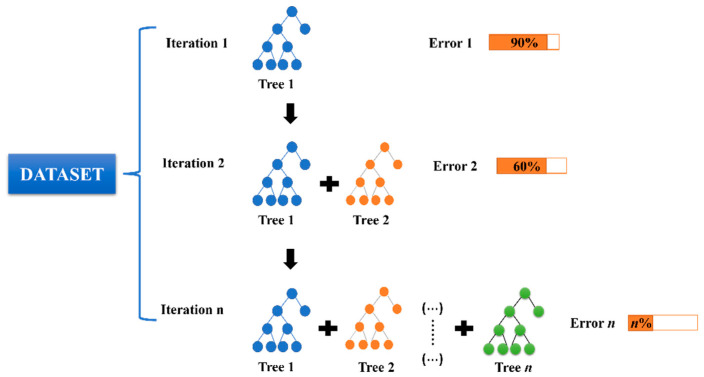
Gradient boosting regression modelling framework.

**Figure 3 materials-17-01192-f003:**
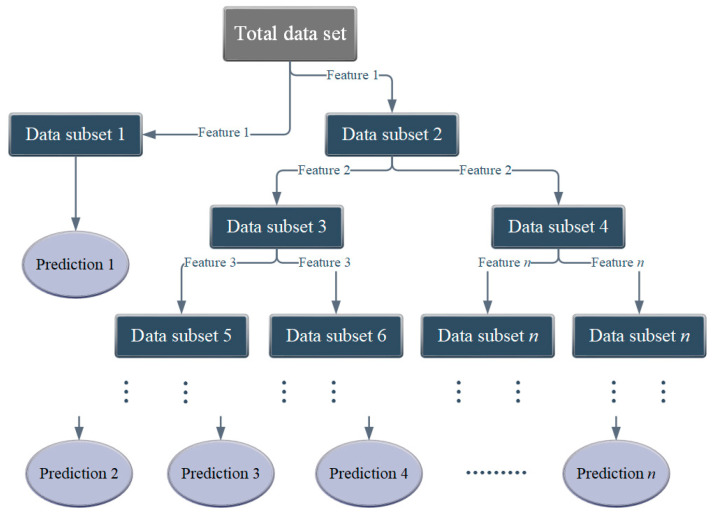
Decision tree modelling framework.

**Figure 4 materials-17-01192-f004:**
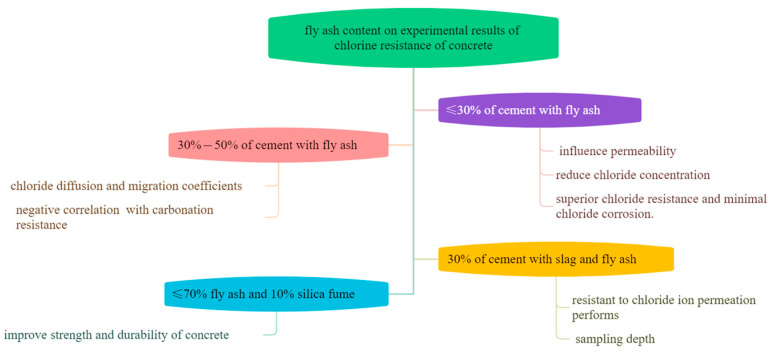
Relationship of fly ash admixture on chlorine resistance of concrete.

**Figure 5 materials-17-01192-f005:**
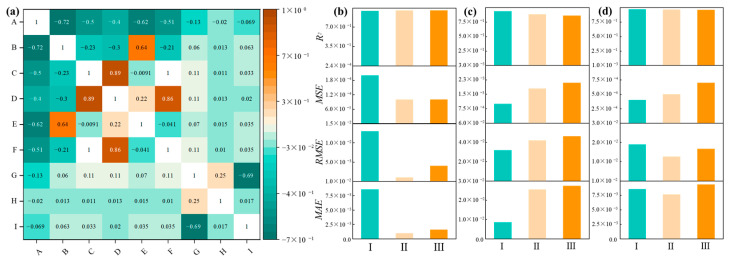
(**a**) Heat map of Pearson correlation coefficient matrix between features (from A to I: cement, fly ash, fine aggregate, coarse aggregate, age, water-binder ratio, depth, soak time, and total chloride content); (**b**) ML model training set, (**c**) testing set, and (**d**) consolidated set (from Ⅰ to ⅠⅠⅠ: RF, GBR, and DT) evaluating the performance of different ML models by *R*^2^, *MSE*, *RMSE*, and *MAE*.

**Figure 6 materials-17-01192-f006:**
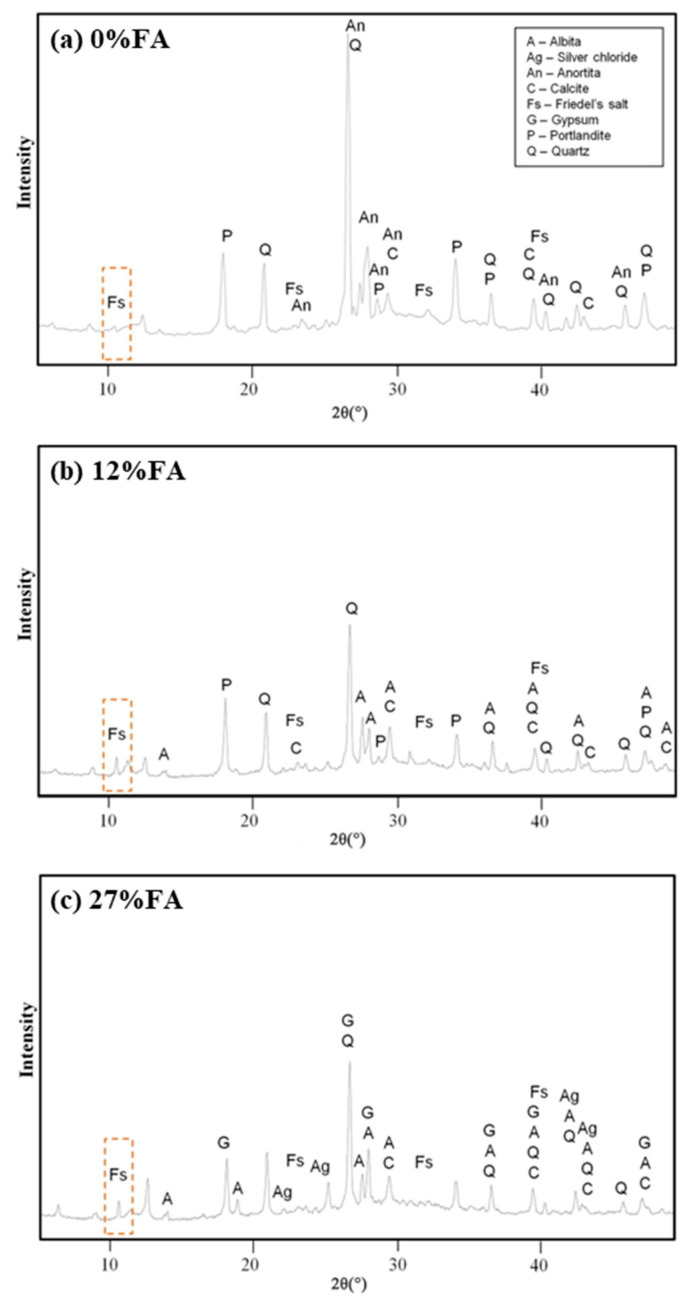
XRD test of concrete with different fly ash admixtures after chloride ion attack, and the results indicate the presence of Friedel’s salt. (**a**) 0% fly ash content, (**b**) 12% fly ash content, (**c**) 27% fly ash content. A reference here [52].

**Figure 7 materials-17-01192-f007:**
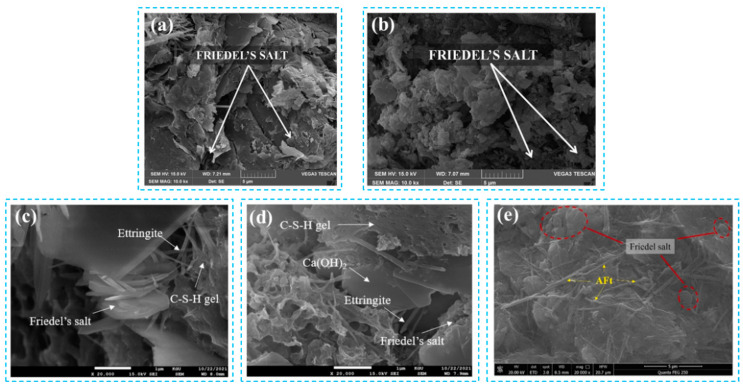
Friedel’s salt in fly ash concrete under SEM. (**a**–**e**) reference here [52,53,54].

**Figure 8 materials-17-01192-f008:**
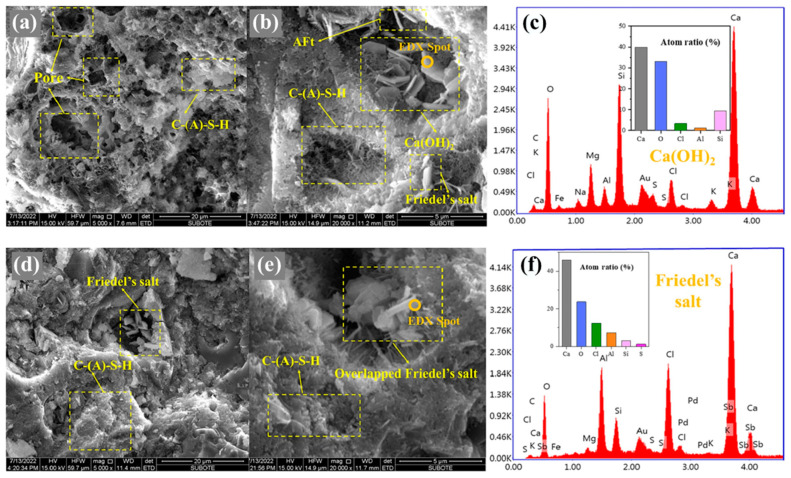
Friedel’s salt in fly ash concrete under EDX. (**a**–**f**) reference here [63].

**Figure 9 materials-17-01192-f009:**
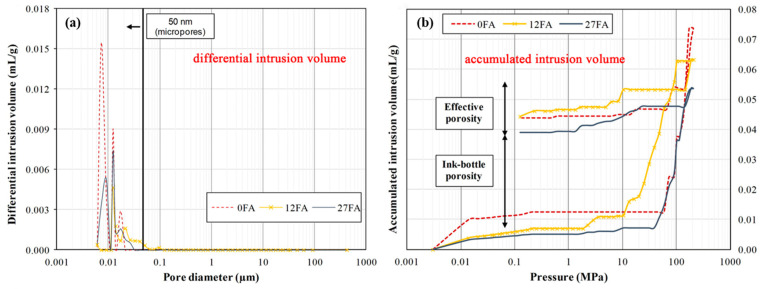
MIP results of concretes after the test: (**a**) differential intrusion volume; (**b**) accumulated intrusion volume. A reference here [52].

**Figure 10 materials-17-01192-f010:**
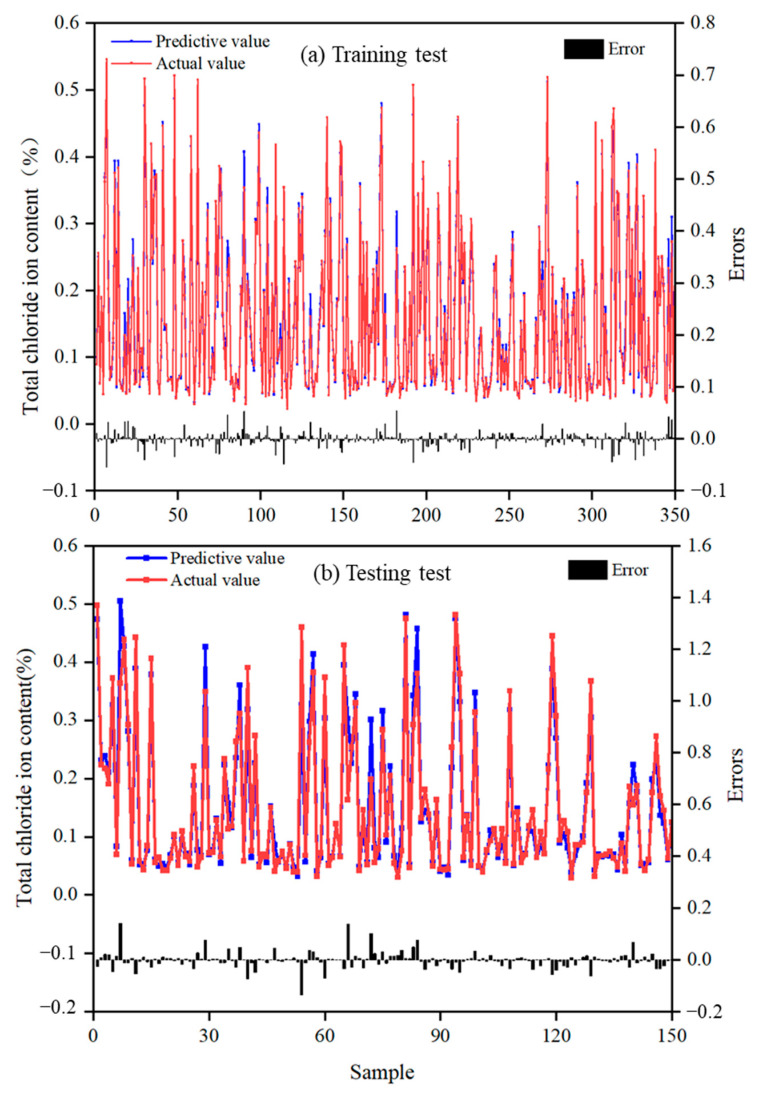
Target and predicted total chloride ion content calculated by RF. (**a**) Training set, (**b**) testing set.

**Figure 11 materials-17-01192-f011:**
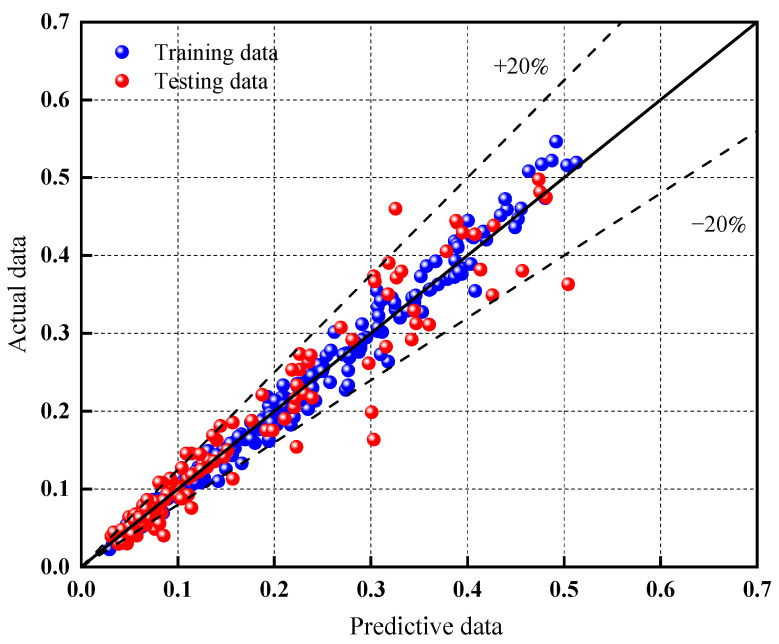
RF model ±20% error distribution.

**Figure 12 materials-17-01192-f012:**
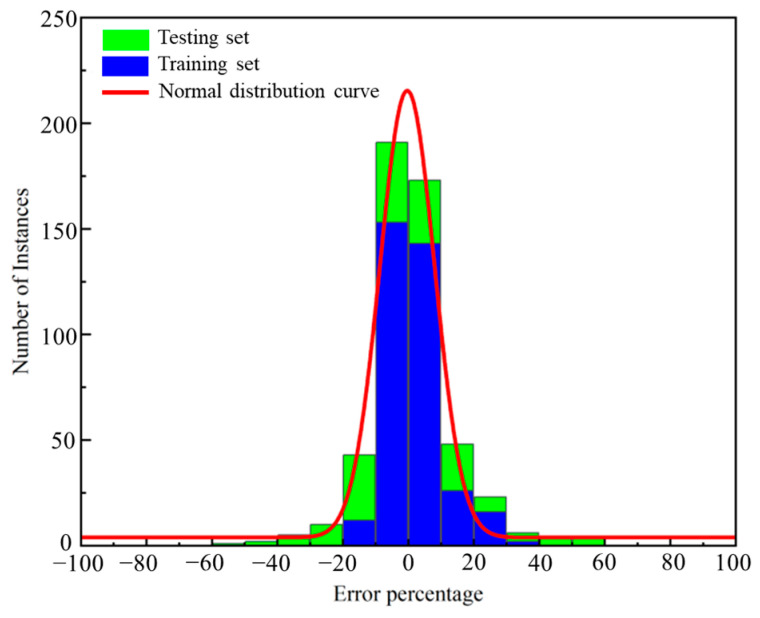
Error percentage distribution and normal distribution fitting of RF model.

**Table 1 materials-17-01192-t001:** Effective and ink-bottle porosity of concretes. A reference here [52].

Pore	0%FA	12%FA	27%FA
Effective porosity	40.59%	29.98%	27.28%
Ink-bottle porosity	59.41%	70.02%	72.72%

**Table 2 materials-17-01192-t002:** Mean, standard deviation, and range of model input/output parameters.

Statistical Characteristics	Cement (kg/cm^3^)	Fly Ash (kg/cm^3^)	Coarse Aggregate (kg/cm^3^)	Fine Aggregate (kg/cm^3^)	Age (Days)	Soak Time (Days)	Depth (mm)	Total Chloride Ion Content (%)
Mean	325.26	112.21	1165.25	602.47	37.74	91.24	10.63	0.15495
Std	80.81	73.37	15.76	37.51	13.34	37.46	8.06	0.12508
Min	217	0	1128	531	28	28	0.5	0.02255
Max	433	217	1178	663	56	140	35	0.54617

**Table 3 materials-17-01192-t003:** Prediction accuracy of all models for total chloride ion content in fly ash-based concrete.

Statistical Characteristics	RF			GBR			DT		
	Training Set	Testing Set	All	Training Set	Testing Set	All	Training Set	Testing Set	All
*R* ^2^	0.9887	0.9351	0.9726	0.9999	0.8859	0.9658	0.9999	0.8632	0.9589
*MSE*	0.0002	0.001	0.0004	0.0001	0.0018	0.0005	0.0001	0.0021	0.0007
*RMSE*	0.013	0.032	0.019	0.0001	0.0423	0.0127	0.004	0.0464	0.0167
*MAE*	0.0086	0.0086	0.0086	0.0001	0.0256	0.0077	0.0016	0.0274	0.0094

## Data Availability

The machine learning model dataset for this paper was collected from the dissertation by Sun, Congtao “Study on durability and life prediction of concrete based on chloride ion erosion”.

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
