# Peer review of "Machine Learning Method to Explore the Correlation between Fly Ash Content and Chloride Resistance"

_materials, 2024, doi:10.3390/ma17051192_

Round 1

Reviewer 1 Report

Comments and Suggestions for Authors

The article explores the use of fly ash to improve the properties of concrete and investigates the potential of machine learning for accurate and efficient prediction of concrete properties.

In the introduction to the article, it would be appropriate to add:
- novelty of your research
- contribution to the scientific community
- an overview of the article's structure
The authors should complete the discussion section, where they write in a critical and balanced manner:
- Summary of main findings
- Interpretation and comparison with existing knowledge
- Implications and impact
- Limitations and future research
Visually, it is necessary:
- improve the readability of Figure 5.
- complete the missing abstract
- strictly adhere to the mdpi template

Reviewer 2 Report

Comments and Suggestions for Authors

I have reviewed this article about Machine Learning methods to explore the importance of Fly Ash content. 

I consider that the manuscript is quite interesting in its scope and objectives as it provides with valuable insights for materials science and industry.

Regarding the English language, I do not see particular problems in the article.

We are given sufficient information about the kind of processes in which the methods proposed can be applied.

The chemical process is logic and correctly followed, through the finding of appropriate correlations.

From the technical point of view, the data presented are sufficient and the graphs are descriptive and coherent.

On the whole, I recognise that so far, the research presented is almost complete.

I acknowledge, that the authors have worked thoroughly on the matter and I suggest that their article could be published with minor amendments.

Summary of evaluation: The article is logically developed from the theoretical and technical points of view; it provides the reader with valuable insights. It can be duly considered for publication after minor polishing.

Author Response

Thanks for your suggestion. However, we do invite a friend of us who is a native Englishspeaker from the USA to help polish our article. And we hope the revised manuscript couldbe acceptable for you.

Reviewer 3 Report

Comments and Suggestions for Authors

In the current manuscript authors described the application of machine learning methods towards exploration between fly ash content and chloride anions resistance. In the light of modern materials engineering, I find the topic very interesting. This manuscript should add value to the existing state of knowledge in the literature.

As I mentioned above, I consider the topic relevant in the field. An extensive theoretical introduction introduces the reader to the subject matter, and the research methodology described does not contain any substantive errors. The manuscript was prepared with great care for its graphic design, the drawings, tables and charts are careful from an editorial point of view. The considerations were very well summarized in the conclusions, and the cited references were properly and skillfully selected. The whole is a solid study, which I recommend for publication in Materials in its current form.

Author Response

We are very grateful to the reviewers for their positive comments on our work!

Round 2

Reviewer 1 Report

Comments and Suggestions for Authors

The authors processed my comments. In my opinion, the manuscript can be published. Need to adjust the numbering of the pictures before posting and check the English.

Comments on the Quality of English Language

The authors processed my comments. In my opinion, the manuscript can be published. Need to adjust the numbering of the pictures before posting and check the English.
